# Advancements in Methods of Classification and Measurement Used to Assess Tooth Mobility: A Narrative Review

**DOI:** 10.3390/jcm13010142

**Published:** 2023-12-27

**Authors:** Gi Youn Kim, Sunjai Kim, Jae-Seung Chang, Se-Wook Pyo

**Affiliations:** Department of Prosthodontics, Gangnam Severance Dental Hospital, Yonsei University College of Dentistry, Seoul 06273, Republic of Korea; gyyeon0317@yuhs.ac (G.Y.K.); sunjai@yuhs.ac (S.K.); jschang@yuhs.ac (J.-S.C.)

**Keywords:** tooth mobility, displacement, diagnostic evaluation, classification, measurement

## Abstract

Evaluating tooth mobility is clinically significant, not only for diagnosing periodontal tissues but also in determining the overall periodontal treatment plan. Numerous studies related to tooth mobility have been conducted over the years, including the proposal of various classifications as well as the development of electronic devices for objective measurement. However, there is still no consensus on the measurement methods and criteria for assessing tooth mobility. In this study, we provide a comprehensive review of past and current tooth mobility classification and measurement methods. In order to propose a new method to intuitively evaluate tooth mobility based on previous studies, a digital approach capable of recording tooth micromovements induced by dynamic load should be considered.

## 1. Introduction

Tooth mobility refers to how loose a tooth is from the alveolar socket. The factors that influence tooth mobility include the height of the supporting alveolar bone, the width of the periodontal ligament, the presence of inflammation, the shape of the root(s), and the number of roots [1,2,3,4,5,6].

Tooth mobility is divided into physiological and pathological categories. In the morning, physiological mobility is at its greatest in all teeth, but this diminishes throughout the day [7]. Individuals with healthy tissue conditions typically exhibit lower mobility compared to those with parafunctional habits [8,9]. Pregnancy primarily leads to physiological changes that are associated with increased mobility, and prolonged unilateral dental function can contribute towards heightened mobility [10]. Pathological mobility refers to a progressive increase in tooth mobility and can be caused by a variety of factors, such as the progression of periodontal disease, loss of the supporting alveolar bone, bruxism, occlusal trauma, root pathology, and pulp inflammation [11,12,13,14]. Pathological tooth mobility arises from quantitative and/or qualitative changes in the tooth’s supporting structures. Tooth mobility can be categorized into two stages: the initial/intrasocket stage and the secondary stage. The initial/intrasocket stage takes place within the periodontal ligament and is attributed to viscoelastic distortion of the periodontal fluid, periodontal fibers, and interbundle content. This stage typically involves movement ranging from 50 to 100 μm under a 100 lb load. On the other hand, the secondary stage results from the elastic deformation of the alveolar bone in response to increased horizontal forces [15]. Tooth mobility is a useful clinical indicator of the biophysical state of the tooth-supporting structures. Therefore, it is essential for the diagnosis of a patient, and it plays a clinically significant role in various dental treatments, including prosthodontics, orthodontics, periodontics, and dental traumatology [16,17,18,19,20,21]. The most commonly used clinical method to assess tooth mobility involves applying pressure to the tooth using two metal instruments or one metal instrument and fingers [22]. The results are indicated by the range of horizontal or vertical displacement of the tooth [23]. Horizontal tooth mobility refers to the degree to which the tooth can move buccally or lingually within the alveolar socket. The evaluation is performed by placing the handles of the dental instruments on either side of the tooth’s mesiodistal axis and applying moderate pressure alternately, assessing the handle of the other instrument [24]. Vertical tooth mobility refers to the degree to which the tooth can move downward within the alveolar socket. This is evaluated by applying pressure to the tooth’s occlusal or incisal surface using the tip of the instrument handle [25]. This method is widely accepted in clinical routines due to its speed and ease of execution. However, it has the disadvantage of subjective interpretation influenced by the clinical sensitivity and perception of each individual, lacking objective confirmation of tooth mobility and reproducibility of results without addressing causal factors [10,26].

Various electronic devices have been developed over the years to objectively measure the degree of tooth mobility [25,26,27]. Although many techniques and devices exist, their reliability for measuring tooth mobility is still limited and ambiguous. This is because it is challenging to accurately replicate tooth displacement influenced by the viscoelastic properties of the periodontal ligament and the impact of the modulus of elasticity under load conditions in laboratory studies [27]. Additionally, both pathological and physiological factors can influence tooth mobility, and different types of teeth exhibit varying ranges of mobility in different individuals, making it difficult to establish precise criteria for tooth mobility measurement methods [28]. Therefore, this study aims to review past and current classification methods and measurement techniques for tooth mobility and investigate the improved research methodologies that can be applied to natural tooth mobility compared to the existing methods.

## 2. Materials and Methods

To find information for this narrative review, electronic searches were conducted for work published from 1951 to 2022 in the following databases: PubMed, Web of Science, and Google Scholar. Article research was performed using the following terms: tooth mobility, tooth displacement, measurement of tooth mobility, evaluation of tooth mobility, and devices for tooth mobility. The search was limited to peer-reviewed publications that were indexed as articles or reviews. The arbitrary inclusion criteria for the articles in this review included the finite element analysis of stress in the periodontal ligament and the clinical implications of tooth mobility. The exclusion criteria were articles not written in English.

This paper focused on the consideration of tooth mobility related to classification and measurement factors for objective diagnosis rather than orthodontic tooth movement. A total of 299 articles were identified with the initial search strategies. Title and abstract evaluations resulted in the deletion of 199 papers due to their irrelevance to the topic. Additionally, four papers were excluded as their main content was not written in English. Consequently, a total of 96 papers were included in the literature review.

## 3. Classifications of Tooth Mobility

In a routine clinical examination, tooth mobility is assessed by immobilizing the tooth between the metallic handles of two instruments and moving it in the buccolingual or buccopalatal direction. Typically, mobility is classified using the Miller index. The Miller index categorizes tooth mobility into four grades. Grade 0 indicates no mobility, where the teeth are firmly stable within their sockets and no movement is detectable during examination. In Grade 1, there is slight mobility, with minor horizontal or lateral movement of the tooth observed. Grade 2 represents moderate mobility, with more noticeable horizontal movement and possible vertical or axial displacement. Finally, Grade 3 signifies severe mobility, where the tooth exhibits significant movement, both horizontally and vertically, often referred to as “floating tooth” mobility [29,30,31].

The classifications of tooth mobility that have been proposed to date are shown in Table 1. All of the classifications in Table 1 commonly define scores and grades for normal mobility and then classify mobility into scores and grades for movements of approximately 1–2 mm. At the highest level of mobility, it encompasses both horizontal and vertical movements. In addition to the classification based on horizontal and vertical tooth mobility, there are other classifications, such as the Glickman classification, which divides mobility into physiologic and pathologic [32]; the Ramfjord classification, which distinguishes mobility based on the extent of normal function [33]; and the Perlitsch classification, which classifies mobility based on the percentage of periodontal support loss [20]. These classifications have the advantage of not only considering simple tooth movement but also actively reflecting the patient’s periodontal condition. However, compared to the classification based on horizontal and vertical mobility, these classifications have much more subjective judgment criteria and struggle to quantify tooth mobility, which limits their clinical use [20].

## 4. Devices for Tooth Mobility Measurement

Along with the classification methods for tooth mobility, various devices have been developed to evaluate tooth mobility in a more objective manner. In the early stages of device development, a static loading method was utilized to measure tooth mobility by applying force and visually assessing the displacement of the tooth. This method involved manually moving the tooth to evaluate its mobility, which was a common approach employed by dentists to assess tooth mobility. However, this method is subjective, has low reproducibility, and presents challenges in achieving precise numerical measurements [34]. Subsequently, a dynamic loading method was developed to measure tooth mobility. This method enabled the accurate measurement and quantification of tooth mobility [10]. With further technological advancements, electronic devices were introduced to measure tooth mobility. These devices apply forces and electronically measure the tooth’s response, enabling precise quantification [31,46,47,48]. They have progressively aimed to achieve accurate and consistent measurements of tooth mobility, and their characteristics are outlined in Table 2.

### 4.1. Displacement Measuring Devices

The first device, Elbrecht’s Indicator, appeared in 1939 [49]. This device uses a static loading method to measure tooth mobility by measuring the labio-lingual displacement generated through digital pressure using a large-dial indicator. It is capable of measuring mobility only above 0.75 mm, and applied force cannot be separately measured. Furthermore, the execution of this technique necessitated a considerable magnitude of force to induce a displacement of 1/1000 inches, thus posing a formidable challenge [50].

Subsequent advancements led to the development of various types of periodontometers [18,30,32,49,50,52]. The utilization of periodontometers required customized clutches or trays, primarily limiting their use for research purposes. Moreover, the need for customized tools, as opposed to standardized ones, indicates the challenge of maintaining consistent and precise control over loading rates and applied loads [53]. Therefore, persistent endeavors have been dedicated to objectively controlling the variables that impact the outcomes [54,56,58,59,61].

Holographic interferometry, an application of laser technology, offered intricate and comprehensive insights through non-contact and non-destructive means. Nonetheless, the intricate nature of the procedure hindered its clinical implementation [60,61].

Both of the methods mentioned above, periodontometers and holographic interferometry, utilized static loading to measure tooth mobility. Periodontometers use a measuring probe that is inserted between the tooth and the periodontal ligament to apply a specific force and measure mobility. Holographic interferometry involves shining a laser beam on the tooth to capture its initial state and record the interference pattern to create an initial hologram. Then, a static load is applied to the tooth, and the laser beam is, again, directed onto the tooth to record the interference pattern and create a hologram of the deformed state after applying the load. By comparing the initial hologram with the deformed state hologram, the difference in tooth movement is analyzed.

Konermann’s novel intraoral measuring device applies dynamic loading to measure tooth mobility. The dynamic loading is exerted on the teeth using the device’s splint, which moves at a constant speed, applying dynamic force to the teeth. As a result, the teeth undergo displacement, and the amount of movement is measured using laser holographic technology. It was introduced to overcome the difficulties associated with the complex configuration and handling of laser holographic technology [62] or manually driven equipment [65,66] in clinical use. The goal of this technology was to reveal characteristic changes in the periodontal ligament during the maintenance period after orthodontic treatment and record the mobility of teeth within the oral cavity. This device has demonstrated high accuracy and effectiveness in practical use. The precise, fine grading of deflection durations has become an indicator of tooth movement. However, a drawback is that the measurement results can vary depending on unwanted movements by the patient and how the investigator applies the splint [47].

The most recent method used for tooth mobility measurement also utilizes the static loading method. Intraoral scanner measurements aim to provide three-dimensional quantitative results regarding tooth movement using an intraoral scanner within the oral cavity. This is a user-friendly, non-invasive technique that eliminates the need for separate devices such as splints, resulting in less variability due to investigator-dependent results. The intraoral scanner is then used to obtain 3D files of tooth positions, which are then analyzed using measurement software. Tooth mobility is then measured by calculating the linear deviations along three axes (x, y, and z) based on three reference points (cervical (C), middle (M), and occlusal (O)) in the interproximal areas [48].

### 4.2. Strain-Measuring Devices

Picton’s Gauge [54] uses resistance wire strain gauges to measure tooth mobility. Strain gauges detect the vertical movement of teeth, measuring displacement or mobility. One end of the gauge is attached to a single tooth, while the other end is connected to adjacent teeth through a spring. The displacement of the test tooth relative to adjacent teeth is detected by two strain gauges. The measurement of each tooth requires the insertion of a custom assembly. Using these custom assemblies, tooth stress and displacement are measured, and this information is recorded through a Wheatstone bridge circuit.

In Persson and Svensson’s transformer [63], strain gauges and a differential transformer are employed to sense force and displacement. Tooth mobility is recorded at the same location and direction as the loading force, and the signals are documented using a two-channel potentiometric recorder.

They utilize strain analysis using sensors distributed on the tooth surface to measure the deformation caused by the forces applied to the tooth. This technique can provide quantitative information about stress distribution and tooth mobility.

### 4.3. Modal Measuring Devices

Modal analysis has emerged as the predominant approach within electronic devices employed for assessing tooth mobility [67]. Modal analysis measures a system’s dynamic characteristics in the frequency domain. In the industrial field, it is used in the design of structures such as automobiles, aircraft, spacecraft, and computers [68]. In the dental field, non-invasive techniques such as damping capacity assessment (DCA) and resonance frequency analysis (RFA) are utilized. Both methods involve the use of controlled force to detect lateral movement and measure stability, but they differ significantly regarding their technical aspects [69]. RFA utilizes the piezoelectric effect to generate deflection in implants, requiring a transducer such as an implant or abutment, making it an unsuitable method for natural teeth [70].

The Periotest value (PTV) is a biophysical parameter that represents the reaction to impact on periodontal tissues [46]. Periotest (Siemens AG, Benssheim, Germany) involves tapping the tooth with a handheld device that applies a tapping load of 8 g at a velocity of 0.2 m/s. The contact time between the tapping load and the tooth is measured by software and converted into PTV. Periotest is suitable for measuring tooth mobility due to its ease of application, ability to measure both horizontal and vertical dimensions, and reproducibility [18,71,72,73,74]. Originally, Periotest was developed to measure the damping characteristics of the periodontal ligament around natural teeth [75,76]. When using Periotest for research, operators have tried to standardize the condition of assessment with several types of positioning jigs [77,78]. However, there are many parameters that could influence mobility and are difficult to control, leading to unstable results. PTV changes linearly with contact time between the tapping head of Periotest and the tooth surface in the lower range (PTV ≤ 13), while it changes quadratically in the upper range (PTV > 13) [79]. It has been suggested that the difference in the change in PTV based on the range might be affected by the resistance change in interstitial or vascular fluids against loading in the early stages of periodontitis [63,80]. Therefore, it is currently primarily used for measuring the mobility of implants [81,82].

A recently developed modified DCA device (Anycheck, Neobiotech Co., Ltd., Seoul, Republic of Korea) has been improved compared to conventional DCA devices by reducing the amount of impact and discontinuing the tapping action when stability is low, thus reducing the impact on teeth and implants [67,83,84].

The DCA method applies repetitive impact to the tooth surface, measuring the tooth’s rebound velocity and direction. Depending on the mobility of the tooth, the rebound velocity and direction will vary. By analyzing the measured rebound information obtained using dynamic loading, the device provides an immediate numerical representation of the tooth’s mobility.

## 5. Current Strategies and Limitations

Despite continuous efforts to refine the existing classifications of tooth mobility [34] and objective measurement methods for multidirectional mobility [35], the establishment of clear measurement techniques and criteria remains ambiguous and lacks consensus [85]. Initially, most methods focused on inducing static force and measuring the resulting tooth displacement using devices such as dial gauges, indicators, dynamometers, and periodontometers. In cases where displacement measurement was challenging, assessing the staining around the tooth was used as an alternative. Before 1983, standardizing tooth mobility measurements using devices was not widespread outside of the research field due to complexity and long operating times [49,52,53,57,86]. Subsequently, the development of the DCA method (Periotest) by Schulte et al. introduced dynamic force instead of static force, measuring contact time to enable simpler and more convenient tooth mobility measurement in clinical settings. However, the DCA method had difficulties reflecting the condition of the periodontal tissue and lacked appropriate judgment criteria, leading to its predominant use in measuring implant mobility [81,82,87]. To address these issues, no-contact vibration devices were proposed to calculate resonance frequency, elastic modulus, and viscosity coefficients using the frequency response characteristics induced by tooth vibration. However, these methods were limited to implants as they required the presence of intermediary elements such as metal pegs [69,87]. More recently, tooth mobility assessment using intraoral scanners and generating continuous variables related to results under load has been proposed. This approach offers user-friendly and non-invasive features, allowing for objective evaluations irrespective of the clinician’s intention. However, it introduces errors during the digitalization process and is expected to be more limited to assisting in interpreting scan results than immediate tooth mobility judgment [88]. Further research is needed to explore the clinical feasibility of this method [47,48].

The first step in designing accurate tooth mobility measurements is to understand the characteristics of the supporting structures around the tooth. The tooth is composed of enamel, dentin, cementum, and pulp, while its supporting structures include the periodontal ligament, gingiva, and alveolar bone. Among them, the periodontal ligament located on the cementum surface plays a vital role in connecting the tooth to the bone and is closely related to tooth mobility [89]. The periodontal ligament consists of cells and an extracellular compartment, which includes ground substance and fibers. It exhibits nonlinearity, viscoelasticity, and heterogeneity [90]. The elasticity of the periodontal ligament absorbs external impacts, protecting both the tooth and alveolar bone. Additionally, numerous cells in this space act as a defense barrier against inflammation [89].

One common method to simulate the complex characteristics of the periodontal ligament is the finite element method, which converts 3D-modeled features of the periodontal ligament into numerical data, allowing quantitative calculations of stress and deformation under various static and dynamic loads [91,92,93,94,95]. This approach enables a comprehensive and accurate evaluation of phenomena occurring within biological structures like the periodontal ligament, including functional reactions in both healthy and pathological states [89]. However, since the periodontal ligament is intricately connected to various components of the stomatognathic system, simulating its mechanical behavior requires complex analysis, and fully incorporating all its properties can be challenging [96]. Therefore, a multifaceted approach is necessary to objectively study tooth mobility.

Over time, the evaluation methods for tooth mobility have continuously evolved, ranging from manual techniques to the use of electronic devices. These advancements have included transitioning from static forces to dynamic forces and attempts to analyze tooth responses to forces in three-dimensional changes, moving beyond two-dimensional displacement. Tooth mobility induced by static force simply reveals the final position of the tooth without continuous information about the entire process of tooth movement. However, if tooth mobility induced by dynamic load is analyzed through change in velocity rather than position, more accurate measurements are expected because various information about the state of the tissue around the tooth can be confirmed. Digital holographic interferometer-based systems for vibration measurements can be used for this process. A laser Doppler vibrometer is described in which holographic optical elements are used to provide the interferometer reference and object illumination beams.

## 6. Clinical Implications and Challenges

If tooth mobility is not measured accurately, it may result in the inability to distinguish between normal mobility and mobility caused by an ongoing disease, leading to the improper formulation of treatment plans. Furthermore, without a proper assessment of tooth mobility, a precise understanding of the tooth’s condition is compromised, potentially causing treatment delays, the exacerbation of progressive issues, and the eventual failure of dental interventions. Therefore, most clinicians agree that the assessment of tooth mobility plays a crucial role in the success of dental treatments [20,21,57].

To establish objective criteria for tooth mobility, it is essential to differentiate between physiological and pathological mobility. Physiological mobility, or normal mobility, usually does not contribute additional risks related to the severity and progression of the disease, making the judgment relatively straightforward. However, pathological mobility, which is related to the treatment plan and prognosis, requires accurate measurement and appropriate judgment by clinicians. According to Miller’s tooth mobility classification commonly used in clinical practice, mild and moderate mobility include only horizontal movement, while severe mobility encompasses vertical movement. Teeth with severe mobility involving vertical movement are defined as “floating teeth” and diagnosed as “hopeless teeth”, leading to extraction, making it relatively easy to diagnose or distinguish them. Clinically, teeth with Miller’s Grade 3 mobility, characterized by vertical displacement, can be assessed easily with simple examinations. However, distinguishing between physiological and pathological mobility or differentiating between Miller’s grades 1 and 2 mobility often involves subjective judgments. Mild and moderate mobility, which includes only horizontal movement, has a significant impact on the overall treatment plan depending on the degree of movement. Hence, it necessitates objective measurement and clear clinical judgment [85].

Therefore, there is a need within this category to develop objective measurement methods and establish new criteria for more accurate assessments. Furthermore, various devices have been introduced for quantitative and objective measurement of tooth mobility, and the challenges include errors during the digitalization process and the need for careful interpretation of measurement results. This highlights the necessity for further clinical investigation and improvement, addressing the challenges in digital processes, and enhancing the reliability of measurement outcomes.

## 7. Conclusions

In this review paper, the authors note that many of the various classification systems cited in the periodontal literature are highly dependent on the individual investigator and are considered subjective. To increase the accuracy and reliability of tooth mobility measurement, large-scale clinical research and data analysis are needed to identify the factors affecting tooth mobility and develop customized evaluation methods that take individual differences into account. Integrating clinical measurement devices with biomechanical modeling and computer simulations can yield more accurate results. This multimodal approach is essential for advancing tooth mobility assessments. However, tooth mobility assessment in clinical settings must be both simple and intuitive. For a more objective evaluation of tooth mobility, the introduction of new measurement methods using dynamic loading and digital scanning technology is necessary.

## Figures and Tables

**Table 1 jcm-13-00142-t001:** Current tooth mobility grading systems.

Index	Miller [29,34,35,36,37,38]	Wasserman [39]	Lovdal [40]	Armitage [41]	Schluger [42]	Grace and Smales [43]	Prichard [44]	Carranza [45]
0	No mobility	N/A	Normal mobility	N/A	Normal mobility	No mobility	N/A	N/A
1	Greater than normal (physiological)	Normal mobility	Greater than normal	<1 mm mobility	<1 mm in buccolingual direction	<1 mm in buccolingual direction	Slight mobility	Slight mobility
2	<1 mm in buccolingual direction	<3/4 mm	Conspicuous mobility in the axial direction	>1 mm but not depressible	<2 mm in buccolingual direction	1 mm–2 mm	Moderate mobility	Moderate mobility
3	>1 mm in buccolingual direction and depressible	3/4 mm–2 mm (>2 mm to score 4, >2 mm and depressible to score 5)	Mobile in axial and transverse directions	>1 mm and depressible	>2 mm in buccolingual and apical direction	>2 mm in buccolingual or vertical direction	Extensive mobility	Severe mobility

**Table 2 jcm-13-00142-t002:** Electronic measuring devices for quantification of tooth mobility.

Proposal	Device	Force	Quntification	Features
Elbrech (1939) [49,50]	Dial indicator	Digital pressure/static	Displacement	(1) Only movements greater than 0.75 mm are measured. (2) Force is not measured.
Werner (1942) [49]	Oscillator	700 g pressure/static	Displacement	(1) Movements less than 0.25 mm cannot be measured. (2) Only a force of 700 g can be applied.
Muhlemann (1951/1954) [49,51,52,53]	Periodontometers (Macro/Micro)	100–1500 g pressure/static	Displacement	Fixed with a/an impression tray/rubber dam clamp.
Picton (1957) [54,55]	Resistance wire strain gauges	20 N/static	Strain	A customized clutch is required.
Parfitt (1960) [36,56]	Inductive transducers	10–1000 g pressure/static	Displacement	Claims an accuracy of 0.001 mm ± 7%.
O’Leary and Rudd (1963) [52,57]	USAFSAM periodontometer	500 g pressure	Displacement	The device is fixed to one arch.
Korber (1967) [58,59]	Inductive transducers	Dynamic/unclear	Displacement	(1) Does not affect the measurement target. (2) Not widely used due to its complex usage.
Wedendal (1974) [60,61,62]	Dental holographic interferometry	2 N/static and dynamic	Displacement	Special retro-reflective paint is required for surface preparation before holography.
Persson and Svensson (1980) [63]	Linear variable differential transformer	20, 50 and 80 g pressure/static	Displacement/strain	(1) Capable of recording both force and displacement in the same direction. (2) Minimal force applied.
Schulte (1992) [46,64]	Periotest	25 N/dynamic	Modal	Reproducible.
Konermann (2017) [47]	Intraoral measuring device	0.05 N–200 N/dynamic	Displacement	Results can vary depending on unwanted movement by the patient and how the investigator applies the splint.
Meirelles (2020) [48]	Intraoral scanner measurements	Subjectivity/static	Displacement	Objective assessment of tooth displacement without the operator’s perception.

## Data Availability

The data that support the findings of this study are available on request from the corresponding author.

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
