# Peer review of "Advancements in Methods of Classification and Measurement Used to Assess Tooth Mobility: A Narrative Review"

_jcm, 2023, doi:10.3390/jcm13010142_

Round 1

Reviewer 1 Report

Comments and Suggestions for Authors

In general, the article is well-crafted, providing a thorough and organized exploration of the subject matter with a comprehensive overview, albeit slightly broader than essential.

One notable observation is the presence of repetitions in the introduction, which is also somewhat lengthy. Specifically, paragraphs 2 and 6 (commencing at lines 30 and 69) contain duplicated information. A suggestion would be to merge these paragraphs into a single cohesive unit, eliminating unnecessary duplication and contributing to a more streamlined and concise introduction.

Author Response

Comment:

In general, the article is well-crafted, providing a thorough and organized exploration of the subject matter with a comprehensive overview, albeit slightly broader than essential.

One notable observation is the presence of repetitions in the introduction, which is also somewhat lengthy. Specifically, paragraphs 2 and 6 (commencing at lines 30 and 69) contain duplicated information. A suggestion would be to merge these paragraphs into a single cohesive unit, eliminating unnecessary duplication and contributing to a more streamlined and concise introduction.

Response: 

Dear reviewer. I sincerely appreciate your positive evaluation of this manuscript. To follow your recommendation, I merged paragraphs 2 and 6 into one paragraph and shortened the content for brevity.

Reviewer 2 Report

Comments and Suggestions for Authors

COMMENTS TO THE AUTHORS

1.     The introduction section is longer than necessary; I believe that the main reason for this is that the authors repeat similar findings. For this reason, I recommend that researchers organize paragraphs properly and present paragraph topics in a sequence that the reader can understand.

2.     Also, it is quite remarkable that the number of references in the introduction section alone is 38. And it indicates that researchers need to simplify this section. Please revise your bibliography by choosing up-to-date references.

3.     I think there is a methodological error in the design of the study. Although the study was not planned as a meta-analysis, it is not appropriate to include some keywords such as periodontal or tooth movement, which are preferred for literature review, due to their secondary relevance considering the research topic.

4.     Additionally, the exclusion criteria of the study are quite inadequate; While irrelevant keywords are already included in the inclusion criteria, it is far from scientific to say that irrelevant articles are not included.

5.     Detailed information should definitely be given in the results about how the results were reduced from 299 articles to 100 articles. How many of them were eliminated due to their irrelevance to the topic and how many were eliminated due to their writing language.

6.     Why did the researchers include traditional book information in their article results instead of the information in the literature? In this case, what is the contribution of the current article to the literature?

7.     It is noteworthy that the researchers included a total of 100 references for the entire article and that the literature used was as old as 1974. In order for the article to contribute to the literature, it definitely needs references based on current literature; Otherwise, it cannot go beyond repeating what is known. Therefore, it may be useful to consider time limitation in exclusion criteria in the methodology.

Comments on the Quality of English Language

Unfortunately, there is a need for extensive editing in terms of English.

Author Response

Dear Reviewers

We thank the reviewers for their comments, and we are grateful for the opportunity to provide further revisions to our paper. We changed our manuscript according to the reviewers’ comments and recommendations. We are trying to adequately address each of the points made by the reviewers. We also corrected extensive English language through English proofreading. We would be very thankful if you could please reconsider a thoroughly revised manuscript. We highlighted the changes made in the manuscript by using a different color font (red): see correction marked form file, and explained details in this letter.

Comment 1:

The introduction section is longer than necessary; I believe that the main reason for this is that the authors repeat similar findings. For this reason, I recommend that researchers organize paragraphs properly and present paragraph topics in a sequence that the reader can understand.

Response 1: 

We followed your recommendations. We have incorporated the qualitative and quantitative aspects of tooth mobility mentioned in paragraph 6 into the content of paragraph 2. This has further reduced the length of the introduction.

Comment 2:

Also, it is quite remarkable that the number of references in the introduction section alone is 38. And it indicates that researchers need to simplify this section. Please revise your bibliography by choosing up-to-date references.

Response 2: 

We followed your recommendations. We excluded some older papers from the introduction, retaining mostly recent ones. As a result, the number of references has been reduced from 38 to 28.

Comment 3:

I think there is a methodological error in the design of the study. Although the study was not planned as a meta-analysis, it is not appropriate to include some keywords such as periodontal or tooth movement, which are preferred for literature review, due to their secondary relevance considering the research topic.

Response 3: 

In materials and methods at first paragraph, we omitted the term “tooth movement”, “periodontal”, and “periodontal ligament”. Additionally, in the abstract keywords, we removed the term 'periodontal ligament.

Comment 4:

Additionally, the exclusion criteria of the study are quite inadequate; While irrelevant keywords are already included in the inclusion criteria, it is far from scientific to say that irrelevant articles are not included.

Response 4: 

We agree with your statement that it is not scientifically sound to claim that irrelevant articles are not included. Therefore, we have excluded the phrase 'not related to the topic.

Comment 5:

Detailed information should definitely be given in the results about how the results were reduced from 299 articles to 100 articles. How many of them were eliminated due to their irrelevance to the topic and how many were eliminated due to their writing language.

Response 5: 

We have added further details on how we selected eligible papers from the 299 papers. Additionally, during the process of modifying the reference list, three papers were excluded, resulting in a total of 96 selected papers.

Comment 6:

 Why did the researchers include traditional book information in their article results instead of the information in the literature? In this case, what is the contribution of the current article to the literature?

Response 6 

Thank you for your favorable comment regarding our papers. The inclusion of traditional books serves two primary purposes in contrast to the literature. Firstly, traditional books provide a comprehensive overview, particularly valuable for the topic of tooth mobility, as they offer a broad perspective that may be challenging to cover in limited pages and space within a paper. Additionally, traditional books contain information about the development and historical evolution of specific fields. This second reason is crucial for understanding the historical context of the topic and providing an overall flow to the research. However, considering that three book references did not significantly contribute to the content, they were excluded.

Comment 7:

It is noteworthy that the researchers included a total of 100 references for the entire article and that the literature used was as old as 1974. In order for the article to contribute to the literature, it definitely needs references based on current literature; Otherwise, it cannot go beyond repeating what is known. Therefore, it may be useful to consider time limitation in exclusion criteria in the methodology.

Response 7: 

The majority of research related to tooth mobility is largely confined to the past, with recent focus shifting towards the stability of implants as a predominant theme. Consequently, it has become challenging to find content specifically addressing natural tooth mobility measurements. Recognizing the need for a new measuring tool in natural tooth mobility assessments, this review article aims to discuss the significance of tooth mobility measurement and advocate for more advanced methodologies. Despite the limited number, we surveyed references up to 2022 to trace the evolution of tooth mobility research from the past to the present. The intention is to provide insights into the direction of future research in this field.

Reviewer 3 Report

Comments and Suggestions for Authors

This manuscript reviews past and current classification methods and measurement techniques for tooth mobility and investigate research methodologies that can be applied to natural tooth mobility. It will help the readers of JCM. However, I have some comments to improve this manuscript. My comments are as follows:

(Line 235-239)

“Originally, Periotest was developed to measure the damping characteristics of the periodontal ligament around natural teeth [83,84]. However, there were many parameters that could influence the mobility and were difficult to control, leading to unstable results. Therefore, it is currently primarily used for measuring the mobility of implants [85,86].”

-When using Periotest for research, operators have tried to standardize the condition of assessment with several types of positioning jig (Nagayama et al., 2020; Uchida et al., 2022). I recommend adding this information here with additional reference as follows:

Uchida, H.; Wada, J.; Watanabe, C.; Nagayama, T.; Mizutani, K.; Mikami, R.; Inukai, S.; Wakabayashi, N. Effect of night dentures on tooth mobility in denture wearers with sleep bruxism: A pilot randomized controlled trial. J. Prosthodont Res. 2022, 66, 564-571. doi: 10.2186/jpr.JPR_D_21_00230.

(Additional Discussion)

In the paper written by Uchida et al., they pointed it out that Periotest has a specific limitation as follows:

“Rosenberg et al. reported that PTV changes linearly with contact time between the tapping head of Periotest® and the tooth surface in the lower range (PTV≤13), while it changed quadratically in the upper range (PTV>13)[Rosenberg et al., 1995]. It has been suggested that the difference in the change in PTV based on the range might be affected by the resistance change of interstitial or vascular fluids against loading in the early stages of periodontitis [Kindlová et al., 1962; Persson et al., 1980].”

This discussion is also helpful for this narrative review, I think. Please check these papers and add some description about the limitation of Periotest if it will be meaningful.

3. Rosenberg D, Quirynen M, van Steenberghe D, Naert IE, Tricio J, Nys M. A method for assessing the damping characteristics of periodontal tissues: goals and limitations. Quintessence Int. 1995;26:191–7. PMID:7568735

4. Kindlová M, Matena V. Blood vessels of the rat molar. J Dent Res. 1962;41:650–60. https://doi.org/10.1177/00220345620410031801, PMID:14032809

5. Persson R, Svensson A. Assessment of tooth mobility using small loads. I. Technical devices and calculations of tooth mobility in periodontal health and disease. J Clin Periodontol. 1980;7:259–75. https://doi.org/10.1111/j.1600-051X.1980.tb01969.x, PMID:6936405

Author Response

Comment 1:

This manuscript reviews past and current classification methods and measurement techniques for tooth mobility and investigate research methodologies that can be applied to natural tooth mobility. It will help the readers of JCM. However, I have some comments to improve this manuscript. My comments are as follows:

 (Line 235-239)

“Originally, Periotest was developed to measure the damping characteristics of the periodontal ligament around natural teeth [83,84]. However, there were many parameters that could influence the mobility and were difficult to control, leading to unstable results. Therefore, it is currently primarily used for measuring the mobility of implants [85,86].”

-When using Periotest for research, operators have tried to standardize the condition of assessment with several types of positioning jig (Nagayama et al., 2020; Uchida et al., 2022). I recommend adding this information here with additional reference as follows:

Uchida, H.; Wada, J.; Watanabe, C.; Nagayama, T.; Mizutani, K.; Mikami, R.; Inukai, S.; Wakabayashi, N. Effect of night dentures on tooth mobility in denture wearers with sleep bruxism: A pilot randomized controlled trial. J. Prosthodont Res. 2022, 66, 564-571. doi: 10.2186/jpr.JPR_D_21_00230.

Response 1: 

Dear reviewer. We sincerely appreciate your positive evaluation of this manuscript and your kind advice. Additional contents and references have been included in the second paragraph under the subheading 'Modal measuring devices.

Comment 2:

(Additional Discussion)

In the paper written by Uchida et al., they pointed it out that Periotest has a specific limitation as follows:

“Rosenberg et al. reported that PTV changes linearly with contact time between the tapping head of Periotest® and the tooth surface in the lower range (PTV≤13), while it changed quadratically in the upper range (PTV>13)[Rosenberg et al., 1995]. It has been suggested that the difference in the change in PTV based on the range might be affected by the resistance change of interstitial or vascular fluids against loading in the early stages of periodontitis [Kindlová et al., 1962; Persson et al., 1980].”

This discussion is also helpful for this narrative review, I think. Please check these papers and add some description about the limitation of Periotest if it will be meaningful.

  1. Rosenberg D, Quirynen M, van Steenberghe D, Naert IE, Tricio J, Nys M. A method for assessing the damping characteristics of periodontal tissues: goals and limitations. Quintessence Int. 1995;26:191–7. PMID:7568735
  2. Kindlová M, Matena V. Blood vessels of the rat molar. J Dent Res. 1962;41:650–60. https://doi.org/10.1177/00220345620410031801, PMID:14032809
  3. Persson R, Svensson A. Assessment of tooth mobility using small loads. I. Technical devices and calculations of tooth mobility in periodontal health and disease. J Clin Periodontol. 1980;7:259–75. https://doi.org/10.1111/j.1600-051X.1980.tb01969.x, PMID:6936405

Response 2: 

Thank you very much for your kind advice. We followed your recommendations. We have added content and references to the second paragraph under the subheading 'Modal measuring devices.

Reviewer 4 Report

Comments and Suggestions for Authors

Review for manuscript ID: jcm-2723067entitled “Advancements in Methods of Classification and Measurement used to Assess the Tooth Mobility: A Narrative Review”

1.       Abstract: the phrase “diagnosing soft and hard tissues of the oral cavity” should be replaced by “periodontal tissues” as there are other soft hard tissues in the oral cavity that are not affected by tooth mobility. And “treatment strategies” should be replaced by the “periodontal treatment plan”

2.       Abstract: there is no conclusion of the study. 

3. The keyword “periodontal ligament” is not related to the topic.

4. Introduction: In line 29, I think the presence of inflammation also affect mobility. 

5. line 79, “scientifically” should be removed. 

6. line 96, the comma after periodontal has to be removed.

7. As this study is a narrative in type, I think there is no need for a section (introduction, methods, results and discussion) like an original study. It has to be sectioned according to the topic itself. 

8. The position of Table 2, has to be after section 3.2.

9. Section 3.2 should be named “Devices for tooth mobility measurement” 

10. line 140 “and scientific manner” should be removed as this means other methods are not scientific. 

11. line 161-166 are not clear, and need to be paraphrased. 

12. section 3.2.1 should be “displacement measuring devices”

13. line 183, the authors mentioned that Konermann's device was used to address the periotest device, yet the periotest device is not mentioned earlier. 

14. Some abbreviations are defined like IOS, PTV and FEM, but not used later, so what is the point of abbreviating? 

15. 3.2.2 should be “Strain measuring devices”

15. 3.2.3. should be “Modal measuring devices”

16. line 226 define RFA. 

17. I think the manuscript in general does not provide enough overview of the topic.

18. there must be a section about the clinical implications and challenges of each index and device. 

19. The authors should highlight the importance of measuring tooth mobility and what happens when it is not measured correctly in term of diagnosis and treatment plan with example.

20. I think the area of incorrect measuring of each of the indices and devices should be compared in a table. 

BW,

Comments on the Quality of English Language

The manuscript needs minor checking for linguistic errors. 

Author Response

Comment 1:

Abstract: the phrase “diagnosing soft and hard tissues of the oral cavity” should be replaced by “periodontal tissues” as there are other soft hard tissues in the oral cavity that are not affected by tooth mobility. And “treatment strategies” should be replaced by the “periodontal treatment plan”

Response 1:

We followed your recommendations. We replaced the phrase “diagnosing soft and hard tissues of the oral cavity” to “diagnosing periodontal tissues” Also, we changed the phrase “treatment strategies” to “periodontal treatment plan”.

Comment 2:

Abstract: there is no conclusion of the study.

Response 2:

We appreciate your comment. We have added the conclusion in the abstract. 

Comment 3:

The keyword “periodontal ligament” is not related to the topic.

Response 3:

We followed your recommendations. We have removed “periodontal ligament” from the keywords.

Comment 4:

Introduction: In line 29, I think the presence of inflammation also affect mobility.

Response 4:

We have added the word 'inflammation' and included one additional reference related to inflammation in the first paragraph.

Comment 5:

line 79, “scientifically” should be removed.

Response 5:

We followed your recommendations. We have removed the word “scientifically” from the fourth paragraph of the introduction.

Comment 6:

line 96, the comma after periodontal has to be removed.

Response 6:

We have removed the word “periodontal”.

Comment 7:

As this study is a narrative in type, I think there is no need for a section (introduction, methods, results and discussion) like an original study. It has to be sectioned according to the topic itself.

Response 7:

Thank you for your favorable comment regarding our papers. We have removed “Result” and “Discussion” and restructuring the sections according to the respective topics.

Comment 8:

The position of Table 2, has to be after section 3.2.

Response 8:

We revised the position of Table 2.

Comment 9:

Section 3.2 should be named “Devices for tooth mobility measurement”

Response 9:

We followed your recommendations. We changed the name of Section 3.2.

Comment 10:

line 140 “and scientific manner” should be removed as this means other methods are not scientific.

Response 10:

We followed your recommendations. We removed “and scientific manner” from the text.

Comment 11:

line 161-166 are not clear, and need to be paraphrased.

Response 11:

We appreciate your comment. We have paraphrased and revised the content to be more clear.

Comment 12:

section 3.2.1 should be “displacement measuring devices”

Response 12:

We followed your recommendations. We changed the name of Section 3.2.1.

Comment 13.

line 183, the authors mentioned that Konermann's device was used to address the periotest device, yet the periotest device is not mentioned earlier.

Response 13:

We apologize for any confusion. We have removed the contents of periotest device that might cause confusion in the flow.

Comment 14:

Some abbreviations are defined like IOS, PTV and FEM, but not used later, so what is the point of abbreviating?

Response 14:

We appreciate your comment. We have reviewed the content and removed any unnecessary abbreviations, but PTV was not removed as it was mentioned in additional content.

Comment 15:

3.2.2 should be “Strain measuring devices”

3.2.3. should be “Modal measuring devices”

Response 15:

We followed your recommendations. We changed the name of Section 3.2.2. & 3.2.3.

Comment 16:

line 226 define RFA.

Response 16:

We appreciate your comment. We applied it in the first paragraph of Section 4.3.

Comment 17:

I think the manuscript in general does not provide enough overview of the topic.

Response 17:

It is difficult to provide an overview of the review papers on tooth mobility because the experimental conditions or results are not standardized for each study. Additionally, these days, because implant mobility studies are more prevalent in mobility measurement, not much research is being done on the mobility of natural teeth. These aspects acted as an obstacle to creating an overview of the topic.

Comment 18:

there must be a section about the clinical implications and challenges of each index and device.

Response 18:

We followed your recommendations. The title “Clinical implications and challenges” related to index and device was added in the manuscript.

Comment 19:

The authors should highlight the importance of measuring tooth mobility and what happens when it is not measured correctly in term of diagnosis and treatment plan with example.

Response 19:

We followed your recommendations. We added content to the title “Clinical implications and challenges” section on the impact of inaccurate measurement of tooth mobility on diagnosis and treatment planning.

Comment 20:

I think the area of incorrect measuring of each of the indices and devices should be compared in a table.

Response 20:

As discussed in this paper, measuring the accuracy of tooth mobility is challenging due to the lack of precise standards, resulting in situations where the accuracy of measurements may be compromised. Judging incorrect measuring objectively in the absence of accurate standards is a difficult task, and there is no precedent for addressing this matter in the literature. Therefore, it is virtually impossible to compare the area of ​​incorrect measuring. Hence, this content has not been included in the table.

Round 2

Reviewer 4 Report

Comments and Suggestions for Authors

NIL